# Evaluation of the virtual learning environment by school students and their parents in Saudi Arabia during the COVID-19 pandemic after school closure

**Moustafa Abdelaal Hegazi**[1,2], **Nadeem Shafique Butt**[3]*, **Mohamed Hesham Sayed**[1,4], **Nadeem Alam Zubairi**[1], **Turki Saad Alahmadi**[1,5], **Mohamed Saad El-Baz**[1,4], **Ali Fahd Atwah**[1], **Mohammad Ahmed Altuwiriqi**[1], **Fajr Adel Saeedi**[1], **Nada Mansour Abdulhaq**[1], **Saleh Huwidi Almurashi**[6]

1 Department of Pediatrics, Faculty of Medicine in Rabigh, King Abdulaziz University, Jeddah, Saudi Arabia, 2 Department of Pediatrics, Mansoura University Children's Hospital, Mansoura, Egypt, 3 Departments of Family and Community Medicine, Faculty of Medicine in Rabigh, King Abdulaziz University, Jeddah, Saudi Arabia, 4 Department of Pediatrics, Faculty of Medicine, Cairo University, Cairo, Egypt, 5 Department of Pediatrics, King Abdulaziz University Hospital, Jeddah, Saudi Arabia, 6 Faculty of Medicine in Rabigh, King Abdulaziz University, Jeddah, Saudi Arabia

* nshafique@kau.edu.sa

**Data Availability Statement:** doi: 10.17632/wrd7xmrdjc.1.

## Abstract

### Background

Very few previous studies have involved school students or their parents in the evaluation of virtual learning environment (VLE). Thus, this survey was performed to evaluate the satisfaction of both school students and their parents with the VLE in the Kingdom of Saudi Arabia during the COVID-19 pandemic.

### Methods

A cross-sectional questionnaire-based survey was distributed online for VLE evaluation. The questionnaire was based on previous studies and expert opinions from validated instruments for assessing distance education, integrative and literature reviews of VLE environment. A median value >3 indicated participant satisfaction in each of the 5 domains of the questionnaire as well as overall VLE satisfaction. The used questionnaire was checked after its implementation by all possible statistical means and it was found to be of acceptable validity and reliability.

### Results

Six hundred and ninety-three participants including 571 Saudi citizens and 122 non-Saudi residents participated in this survey. The number of school students who agreed or strongly agreed were significantly lower than the number of students who disagreed or strongly disagreed with preferring the VLE over traditional education (p<0.001). The participants evaluated the VLE experience as unsatisfactory with a median value ≤3 for 4 out of 5 questionnaire domains with an overall satisfaction value of 2.8. Among the 117 participants who gave further written opinions/comments, 42(35.9%) participants supported the VLE as

**Funding:** The author(s) received no specific funding for this work.

**Competing interests:** The authors have declared that no competing interests exist.

**Abbreviations:** KSA, Kingdom of Saudi Arabia; PASS, Power analysis & Sample size software; SD, Standard deviation; SPSS, Statistical Package for the Social Sciences; VLE, Virtual learning environment.

an alternative to traditional classrooms, if equipment and internet are made available and for the safety of their children.

## Conclusions

This is one of few available adequate population-based studies for exploring the VLE satisfaction of both Saudi citizens and non-Saudi residents school students and their parents. This study showed the participants' unsatisfactory VLE experience. The VLE is accepted as an alternative to traditional classrooms to keep up with learning and to maintain the safety of children and it can be a supplementary learning method but many measures are still needed to develop the VLE.

## Introduction

Much has been reported about the impact of the ongoing COVID-19 pandemic on the health and lives of people throughout the world. As almost every aspect of life was affected, the educational process at every level has also been influenced when governments closed schools and ordered students to stay home [1]. Quick modification of the method of delivering education and of the assessment process was needed. The answer was a complete shift to distant e- learning constituting a virtual learning environment (VLE).

In VLE, remote learners receive their learning materials electronically via the internet [2]. Although the VLE/e-learning has already experienced high growth/development in the past few years as an alternative to traditional classrooms, the dependence on and the use of VLE have increased significantly since the start of the COVID-19 pandemic [3]. Compared to the physical classroom environment, the VLE employs technology to conduct courses and lessons, offer quizzes, and provide various assessment tools [4].

However, the VLE has many disadvantages such as the lack of prompt live interaction and feedback in asynchronous e-learning, the excess time that instructors need for preparation, and potentially more frustration especially in the absence of high-speed internet or the required technical experience or support [2, 4]. Additionally, school students and their caregivers may not have the required skills and experience to adapt and use VLE technology incurring the risk of academic failure of students who are unable to complete their homework and assignments [5].

In response to COVID-19 pandemic, governments implemented lockdown measures with closure of public places and schools in the interest of public health to limit spread of infection. Accordingly, the COVID-19 pandemic obliged governments to adopt and implement the VLE with all its advantages and disadvantages as billions of school students around the world had to undergo a sudden unplanned shift from traditional classrooms to home education through the VLE with due support from many concerned organizations [3, 6].

School closure during the COVID-19 pandemic deprives children from essential learning and opportunities for adequate growth and development. Home confinement will have deleterious effects in terms of long-term socioemotional imbalance [7]. The cessation of traditional education is expected to disintegrate the sense of normality that used to be provided by schools. Some children develop socioemotional difficulties, health challenges and learning delays because of home confinement [8]. Moreover, with school closure, parents are more responsible for facilitating the home learning of their children and they usually struggle to perform this task. This is particularly evident for parents with limited education and resources

[9]. Thus, parents, psychologists, governments, non-governmental organizations and in particular pediatricians are very concerned about alleviating the harmful effects of the COVID-19 pandemic on the health/development of children and adolescents, who are the most vulnerable to many of the devastating effects of COVID-19.

The higher education institutions in developed countries were partially utilizing the VLE even before the pandemic. They were equipped with the required skills/experience to embrace this sudden change [2, 10, 11]. This may not be the case for e-learning at the school level as students, caregivers, teachers and institutions may be deficient either in skill or equipment or both. In the past, the use of VLE in the Kingdom of Saudi Arabia (KSA), was mainly oriented towards university level education whereas its use at the school level remained limited [12–14].

To the best of our knowledge, very few previously published studies and none from the KSA have involved both school students and their parents in the evaluation of the VLE. Therefore, this study was performed to use a validated reliable questionnaire as a screening tool to evaluate the satisfaction of both school students and their parents with the main aspects of the VLE in the KSA during the COVID-19 pandemic after school closure.

## Materials & methods

### Ethics approval and participant consent

This study was approved by biomedical ethics unit of Faculty of Medicine of King Abdulaziz University (Reference No 315–20). The data collection protocol adhered to institutional and national ethical standards. Data anonymity and confidentiality were preserved. A written informed consent was not obtained from a parent or guardian for participants under 16 years old as this survey study utilized an online questionnaire and filling the required questionnaire by participants was considered their consent for participation in this survey.

### Study design and participant selection

A cross-sectional questionnaire-based survey was designed to collect data on the main aspects of VLE. In Saudi Arabia, VLE utilized web-based software systems designed to facilitate learning and teaching with the use of tools and activities for students who can perform tasks in class or at their homes and can be engaged into synchronous or asynchronous discussions to help develop their learning. The assessment process was also conducted via VLE. VLE in schools was conducted by trained teachers who have undergone training programs and workshops for E-learning. However, training period might have been relatively short to allow for quick shift to VLE in face of the rapidly emerging COVID-19 crisis which obliged the government to close schools and order students to stay home. This VLE required parents to monitor and help students especially primary school students during their learning process. This survey was conducted from March-May, 2020 during the only and the whole lockdown period for COVID-19 pandemic in KSA. The participating school students and their parents were included by a convenient sampling design by filling an electronic online questionnaire. Parents were invited to answer this questionnaire if they were Saudi citizens or residents and had school students aged 6–18 years who had gone through the VLE experience. Participants living outside the KSA or having students outside the specified age range of 6–18 years as well as incomplete questionnaires were excluded from this study.

### Measuring tool

A questionnaire based on the main components and basic requirements of e-learning was developed by the researchers as an exploratory screening tool to evaluate the overall

experience/satisfaction of both school students/learners and their parents with the VLE experience and to identify difficulties encountered while using the VLE. The items of the questionnaire suitable for school students were based on previous studies and expert opinions from validated instruments for assessing distance education learning environments in higher education, integrative review for barriers and solutions to online learning and literature reviews and simultaneously included the opinion of parents in VLE [2, 11, 15, 16].

The questionnaire included 3 sections. The first section included a full explanation of the concept, objectives, and benefits of this survey. The second section obtained the socio-demographic data of participating family (Saudi citizen or resident, residence area, number of children, age, gender, school grade of the student, education level and occupation of both mother and father). The third section had 24 questions to evaluate the learner and parental perspective on experience with the VLE. At the end of the questionnaire, the participants were invited to add any further opinions or comments.

Five faculty members experienced in e-learning and conduction of questionnaires and surveys from outside the research team provided their feedback and expert opinion on the content coverage of questionnaire to make the final form of the questionnaire and any amendments were made accordingly. The questionnaire was translated and back-translated (English-Arabic) and checked by two bilingual experts.

In this study, learner and parental VLE satisfaction was the main dependent or outcome variable. A score with a median value above 3 indicated overall satisfaction with the VLE experience as well as satisfaction with each individual domain, while a median value below 3 indicated dissatisfaction [17].

## Statistical validation/reliability analysis of the questionnaire

The used questionnaire was checked after its implementation by statistical means, and it was found to be of adequate and acceptable validity and reliability.

Exploratory factor analysis was conducted for 24 variables (questions) using parallel analysis to determine the number of factors to retain with direct oblimin rotation [18]. According to exploratory factor analysis, the whole questionnaire consisted of 5 main domains:

## Learner domains

1. Perspective on the VLE

2. Skill and interaction

3. Perspective on traditional classrooms

## Parent domains

4. Facilities/support

5. Parental role/perspective on the VLE

The answers to the questions were rated on a 5-point Likert scale from 1–5 (strongly disagree = 1, disagree = 2, true sometimes = 3, agree = 4 and strongly agree = 5). The answers to the questions related to the traditional classroom and facilities/support domains were reverse scored because their answers were in the opposite direction to the answers for all other questions as they expressed the participants' dissatisfaction with some negative aspects/drawbacks of the VLE.

The reliability of all domains as well as the individual domains of the questionnaire was analyzed by Cronbach's alpha test and overall reliability of the questionnaire was excellent (Cronbach alpha coefficient = 0.92).

The questionnaire used and technical details of statistical validation and reliability analysis of the questionnaire are given as S1 File.

## Questionnaire implementation and distribution

Both English and Arabic questionnaires were converted into Google Forms, so that participants cold choose to complete the questionnaire that was more convenient for them. Then, the links for both questionnaires were sent via social media such as WhatsApp's, Twitter, and Facebook to the participants.

The questionnaire was directed to the parents of school students who could complete the whole questionnaire, including responses to questions directed to their children after first taking opinions and answers from their eldest children. Furthermore, the parents could use the questionnaire link again to fill out questionnaire(s) for other students between 6–18 years of age. The student/learner was also allowed to directly answer learner-related questions if desired, and if he/she understood the questions with or without help from his/her parents.

## Sample size and study power

The collected cross-sectional sample size of 693 was selected to achieve 91% power to detect a difference of 0.3 between the null hypothesis mean satisfaction of 3.0 and the alternative hypothesis mean of 3.3 with a significance level (alpha) of 0.05 using a two-sided test carried out with Power Analysis & Sample Size (PASS) software (provided in S1 Data).

Additionally, as the main objective of the study was to explore the satisfaction about VLE, the sample size was also estimated based on mean satisfaction score. A common rule of thumb for determining sufficient sample size is the use of the ratio between overall sample size to the number of free parameters estimates with a recommended ratio of about 20 to 1 [19]. The participant to item ratio for this study was approximately 29: 1, where sample size was 693 and the number of variables included was 24. Accordingly, the given sample size was sufficient to produce reliable results.

## Statistical analysis

The data were analyzed by the Statistical Package for the Social Sciences (SPSS) version 25.0 (IBM corporation, Armonk, NY) after checking for completeness and inconsistencies. The data were scrutinized and double-checked before and after entry into the SPSS program. Frequencies and percentages were used to present categorical variables while the mean, standard deviation (SD) and range were used for quantitative variables. The Kolmogorov Smirnov test was used to test data normality. The Kruskal Wallis test was used to test the differences between the median satisfaction values of the studied variables. P values $<0.05$ were considered significant.

## Results

Six hundred and ninety-three participants, including 571 Saudi citizens and 122 non-Saudi residents from 5 main regions of the KSA completed the questionnaire. The participants were parents of 343 (49.5%) boys and 350 (50.5%) girls, with a mean and SD of 12.6±3.8 years of age for all included school students. The participating parents included 167 (24.1%) fathers with a mean age and SD of 44.5±8.3 years and an age range from 23.0–78.0 years; and 526 (75.9%)

mothers with a mean age and SD of 39.0±7.1 years and an age range from 20.0–60.0 years. Four hundred and ten participants (59.2%) had 3 or more children. A high proportion of fathers (73.8%) and mothers (77.5%) had university and postgraduate higher education. Regarding job status, 624 (90%) of fathers and 350 (50.5%) of mothers were employed. The sociodemographic data of the participants are presented in Table 1.

In one direct question of this questionnaire to express preference for the online VLE over traditional education, the number of school students who agreed or strongly agreed (n = 158, 22.8%) was significantly lower than the number of children who disagreed or strongly disagreed (435, 62.8%) on preferring the VLE over traditional education (p<0.001).

According to the definition of satisfaction, participants evaluated the VLE experience as unsatisfactory with a median value of 3 or less for all domains except the parental role domain

**Table 1. Sociodemographic characteristics of participants.**

| Character | | n | (%) |
|---|---|---|---|
| **Nationality** | Saudi citizen | 571 | (82.4) |
| | Non-Saudi resident | 122 | (17.6) |
| **Region** | Central | 110 | (15.9) |
| | Western | 530 | (76.5) |
| | Eastern | 22 | (3.2) |
| | North | 9 | (1.3) |
| | South | 22 | (3.2) |
| **Number of children in family** | 1 | 112 | (16.2) |
| | 2 | 171 | (24.7) |
| | 3 | 183 | (26.4) |
| | 4 | 113 | (16.3) |
| | 5 | 76 | (11.0) |
| | >5 | 38 | (5.5) |
| **School student gender** | Male | 343 | (49.5) |
| | Female | 350 | (50.5) |
| **School level** | Primary (Mean ± SD of age:9.20±1.98, Range:6–14) Years | 328 | (47.3) |
| | Intermediate (Mean ± SD of age:13.80±1.12, Range:12–16) Years | 145 | (20.9) |
| | Secondary (Mean ± SD of age: 17.10±0.98, Range: 15–18) Years | 220 | (31.7) |
| **Father education** | Less than high school level | 53 | (7.6) |
| | High school level | 129 | (18.6) |
| | University level (Bachelor's degree) | 367 | (53.00) |
| | Postgraduate (Master, MD degree) | 144 | (20.80) |
| **Father job status** | Employed full time | 497 | (71.7) |
| | Employed part time | 56 | (8.1) |
| | Unemployed | 69 | (10.0) |
| | Self-employed/private business | 71 | (10.2) |
| **Mother education** | Less than high school level | 40 | (5.8) |
| | High school level | 116 | (16.7) |
| | University level (Bachelor's degree) | 432 | (62.3) |
| | Postgraduate (Master, MD degree) | 105 | (15.2) |
| **Mother job status** | Employed full time | 284 | (41.0) |
| | Employed part time | 48 | (6.9) |
| | Unemployed | 343 | (49.5) |
| | Self-employed/private business | 18 | (2.6) |

**Table 2. Degree or level of satisfaction with the VLE experience.**

| Domain | Median | Minimum | Maximum |
|---|---|---|---|
| Learner perspective on VLE experience | 3.0 | 1 | 5 |
| Learner skill/interaction | 3.0 | 1 | 5 |
| Learner perspective on traditional classroom | 1.5 | 1 | 5 |
| Parental perspective on facilities/support | 2.8 | 1 | 5 |
| Parental role/perspective on the VLE experience | 3.3 | 1 | 5 |
| Overall satisfaction with VLE experience | 2.8 | 1 | 5 |

(median value of 3.3). Additionally, the overall evaluation revealed dissatisfaction with a median score of 2.8 (Table 2).

Except for the facilities/support domain, Saudi citizens had significantly more satisfaction with all domains and more overall satisfaction than non-Saudi residents (P values<0.05, Table 3, Fig 1). A comparison of overall VLE satisfaction between Saudi citizens and non-Saudi residents by child gender, study grade, paternal and maternal education is shown in Fig 1.

Secondary school students demonstrated significantly more satisfaction with the skill/interaction domain than primary and intermediate school students (P = 0.02). Regarding the parents' education level, fathers and mothers with a university level and above demonstrated significantly higher satisfaction with the facilities/support domain than fathers and mothers with less than a university level of education (P = 0.0001, Table 3).

Regarding further opinions/comments at the end of the questionnaire, 117 (16.9%) participants provided their comments. From these comments, the participants could be clearly identified as opponents or supporters of the VLE. There were significantly more opponents of the VLE than supporters: 75 (64.1%) opponents versus 42 (35.9%) supporters ($X^2$ = 18.6, p< 0.0001). The main given reasons for opposing the VLE, in a descending order, were missing social skills and face-to-face interaction which influence normal child social development and instill values/morals (N = 17), technical problems in the availability of equipment and internet (N = 12), working parents who cannot supervise/help their young primary school children (N = 9), the requirement of more time and effort (N = 8) and other infrequent issues (N = 10). It is noteworthy that 19 more opponents did not give reasons for this opinion. In contrast, the supporters mainly accepted the VLE as an alternative to traditional classrooms in emergency situations and thought that it could be just a supplementary learning method (N = 21), if equipment and internet are made available (N = 12) and for the safety of their children (N = 9).

## Discussion

The long-term serious impacts of COVID-19 pandemic on children worldwide are likely to be destructive, with unprecedented risks to the rights and development of all children despite the less severe symptoms and lower mortality rates in children who are infected with COVID-19 compared to other age groups [20]. Children of all ages and in all countries especially low-income countries are at extremely high risk of significant psychological, socioeconomic impacts and learning problems, particularly if the battle to contain SARS-CoV-2 virus is prolonged [21].

To mitigate the spread of SARS-CoV-2 among children, considering that their safety is a priority, the governments of 188 countries introduced social distancing and lockdown measures with unprecedented worldwide closure of face-to-face child services such as schools

**Table 3. Comparisons of satisfaction in each domain and overall satisfaction with the VLE by nationality, study grade, father's and mother's education.**

| | Categories | | | | | P value[a] |
|---|---|---|---|---|---|---|
| **Variable** | **Median, interquartile range (Q1-Q3)** | | | | | |
| | | **Saudi (n = 571)** | **Non-Saudi (n = 122)** | | | |
| **Nationality** | Learner perspective on the VLE | 2.80(2.20–3.80) | 2.40(2.00–3.40) | | | 0.0001 |
| | Learner skill/interaction | 3.12(2.38–3.75) | 2.88(2.25–3.39) | | | 0.01 |
| | Learner perspective on traditional classroom | 1.50(1.00–2.00) | 1.25(1.00–1.75) | | | 0.0001 |
| | Facilities/Support | 2.75(2.00–3.50) | 3.00(2.00–3.50) | | | 0.69 |
| | Parental role/ perspective on the VLE | 3.33(2.67–4.00) | 3.00(2.33–3.67) | | | 0.001 |
| | Overall satisfaction | 2.79(2.29–3.33) | 2.50(2.20–3.00) | | | 0.0001 |
| **School level** | | **Primary (n = 328)** | **Intermediate (n = 145)** | **Secondary (n = 220)** | | |
| | Learner perspective on the VLE | 2.80(2.00–3.60) | 2.80(2.20–3.80) | 2.80(2.20–3.80) | | 0.59 |
| | Learner skill/interaction | 2.88(2.25–3.50) | 3.12(2.38–3.75) | 3.25(2.38–3.88) | | 0.02 |
| | Learner perspective on traditional classroom | 1.25(1.00–2.00) | 1.50(1.00–2.25) | 1.50(1.00–2.25) | | 0.06 |
| | Facilities/Support | 3.00(2.25–3.50) | 2.75(2.00–3.50) | 2.62(2.00–3.50) | | 0.07 |
| | Parental role/ perspective on the VLE | 3.33(2.67–4.00) | 3.33(2.67–4.00) | 3.67(2.67–4.00) | | 0.09 |
| | Overall satisfaction | 2.71(2.25–3.17) | 2.79(2.21–3.35) | 2.85(2.29–3.40) | | 0.23 |
| **Father's Education** | | **Less than high school (n = 53)** | **High school (n = 129)** | **Bachelor's (n = 367)** | **Master and above (n = 144)** | |
| | Learner perspective on the VLE | 2.80(2.13–4.00) | 2.80(2.00–3.80) | 2.80(2.20–3.60) | 2.80(2.00–3.80) | 0.45 |
| | Learner skill/interaction | 3.37(2.38–4.00) | 3.25(2.25–3.75) | 3.00(2.25–3.62) | 3.12(2.55–3.75) | 0.11 |
| | Learner perspective on traditional classroom | 1.75(1.00–2.25) | 1.50(1.00–2.25) | 1.25(1.00–2.00) | 1.25(1.00–2.00) | 0.02 |
| | Facilities/Support | 2.50(1.75–3.33) | 2.50(2.00–3.25) | 2.75(2.00–3.50) | 3.00(2.25–3.75) | 0.0001 |
| | Parental role/ perspective on the VLE | 3.67(3.00–4.33) | 3.67(2.67–4.00) | 3.33(2.67–4.00) | 3.33(2.67–4.00) | 0.09 |
| | Overall satisfaction | 2.96(2.35–3.50) | 2.75(2.19–3.33) | 2.67(2.25–3.24) | 2.83(2.42–3.38) | 0.13 |
| **Mother's Education** | | **Less than high school (n = 40)** | **High school (n = 116)** | **Bachelor's (n = 432)** | **Master and above (105)** | |
| | Learner perspective on the VLE | 2.60(2.00–3.72) | 2.70(2.00–3.60) | 2.80(2.08–3.60) | 3.00(2.20–3.80) | 0.18 |
| | Learner skill/interaction | 3.25(2.35–3.90) | 3.12(2.30–3.75) | 3.00(2.25–3.62) | 3.25(2.75–4.00) | 0.02 |
| | Learner perspective on traditional classroom | 1.62(1.25–2.25) | 1.38(1.00–2.00) | 1.50(1.00–2.00) | 1.25(1.00–2.00) | 0.17 |
| | Facilities/Support | 2.50(1.75–3.00) | 2.50(2.00–3.25) | 2.75(2.00–3.50) | 3.00(2.50–3.75) | 0.0001 |
| | Parental role/ perspective on the VLE | 3.67(2.67–4.53) | 3.33(3.00–4.00) | 3.33(2.67–4.00) | 3.33(2.56–4.00) | 0.30 |
| | Overall satisfaction | 2.83(2.43–3.32) | 2.69(2.21–3.29) | 2.71(2.25–3.25) | 2.92(2.44–3.39) | 0.09 |

[a] Kruskal-Wallis test.

interrupting education for >90% of the world's students or 1.5 billion children and youth establishing a learning crisis [20, 22, 23]. While more than two-thirds of countries have introduced a national distance learning platform, only 30% of low-income countries have done so [24, 25]. The COVID-19 pandemic implies limited or no education for many children. Not only did school closure in many countries with planned extended lockdowns interrupted teaching but many exams have also been postponed, rescheduled or cancelled [1, 23]. The longer schools remain closed with its dramatic major consequences on children, the less likely children will be able to catch up with learning and essential life skills that support a healthy transition to adulthood.

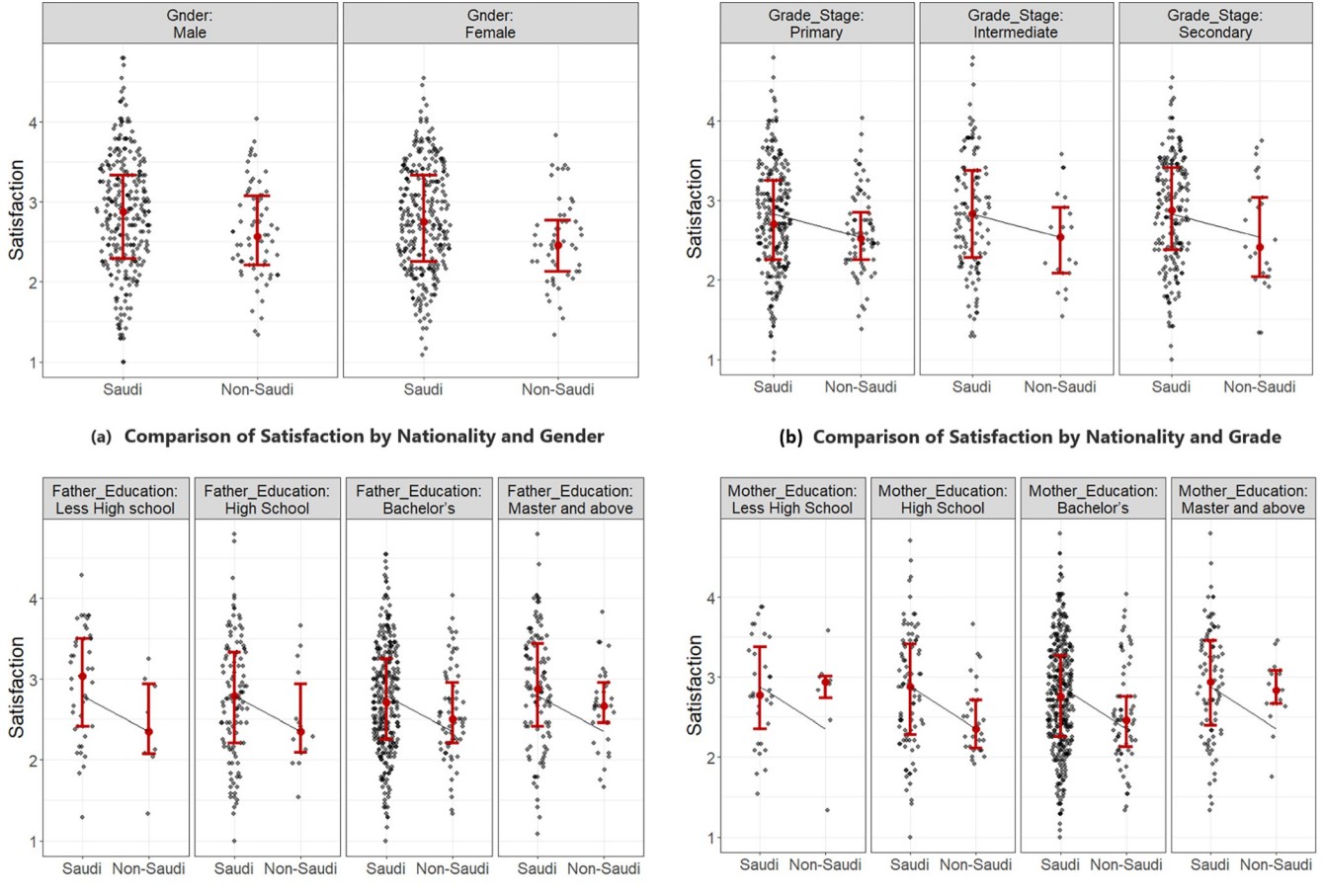

**Fig 1. Comparison of VLE satisfaction by gender, nationality, father's and mother's education level.**

The VLE has been utilized mainly for university students and in the professional e-training of postgraduate students [26] but very few previously published studies, and none from the KSA, have ever involved school students in the evaluation of their experience with the VLE. Therefore, it was initially recognized by the research team that it is essential to use a validated and reliable questionnaire in the evaluation of the satisfaction of school students/learners and their parents with the VLE experience in the KSA after school closure.

In this online survey, 693 participants provided a robust study with a power of 91% that evaluated the VLE satisfaction/experience of the targeted population of primary, intermediate and secondary school children and their parents in 5 main regions of the KSA. The higher participation from the western and central regions of the kingdom can mainly be attributed to the presence of the highest population densities in these 2 regions as well as the uneven distribution of online questionnaires which depended on social media and internet resources/accessibility.

In this survey, the participants evaluated the VLE experience as unsatisfactory with a median value of $\leq 3$ for most of the studied domains and with an overall satisfaction value of 2.8. These results can clearly demonstrate that the participants were not very satisfied with the VLE experience and 62.8% of school children disagreed or strongly disagreed that they preferred the VLE over traditional classrooms. Such a preference for traditional education could

also be recognized from the analysis of the opinions/comments of the participants at the end of the questionnaire as opponents of the VLE were significantly more prevalent than the supporters (p< 0.0001). The most important reasons given by VLE opponents were missing social skills essential for normal child development, problems in the availability of equipment/internet, working parents who cannot supervise/help their young primary school children, the requirement of more time and effort, and more convenience of the VLE for older secondary school students. Parents were only satisfied with their role in supporting the VLE (median = 3.3, Table 2) of their children because it seems that they were convinced and felt that they did their best to facilitate the VLE of their children even though they most likely faced many challenges out of their control and beyond their abilities.

A high proportion of fathers (73.8%) and mothers (77.5%) participating in this survey were highly educated (university level and above) with 90% of the fathers and 50.5% of the mothers being employed. Therefore, they could have more resources for equipment and internet accessibility and a greater ability to respond and participate in this survey. Thus, they were possibly more able to support the VLE of their children as evidenced by significantly more satisfaction of parents with higher education levels with the facilities/support domain compared to parents with lower education levels (p = 0.0001, Table 3). On the other hand, a high proportion of families (59.2%) had 3 or more children and 47.3% of children were in primary schools who may require more supervision/help than older students. Therefore, these families might have experienced a difficult challenge and tough complex situation to allocate time and effort to care and support the VLE of 3 or more children and to provide them with the necessary equipment/internet resources at the same time. Future studies can be more able to thoroughly investigate the various factors that determine parent and child satisfaction including the necessary equipment/internet resources for VLE.

The significantly higher satisfaction values of secondary school students in the skill/interaction domain compared to primary and intermediate school students (Table 3) may be expected because secondary school students are mature enough to be responsible and control their learning and because they have possibly had more training and experience in the use of e-learning technology and the internet. Furthermore, parents expressed that the VLE is more convenient for secondary school students than for primary school students. Therefore, it may seem logical to recommend the VLE mainly for intermediate and secondary school students and school reopening may be resumed initially for primary school children under complete preventive measures given the consistent universal agreement that COVID-19 is less severe in young children than in adults particularly in young children more than 2 years and less than 15 years of age [27–30]. However, other important factors like public health, objective measures of student learning and economic conditions in each country should be considered.

In this survey, Saudi children and their parents had significantly more overall satisfaction with the VLE than non-Saudi residents (Table 3) even though asynchronous e-learning was more commonly applied in Saudi governmental schools than in international schools which mainly serve non-Saudi students. This satisfaction regardless of the somewhat possible lower degree of VLE quality may be related to the greater preference of Saudi children for e-learning with the majority of the Saudi population is actively using the internet and social media [31]. Another explanation may be related to difficulties encountered by expatriate families to adjust to multiple challenges and stressors such as need to readjust life in a new country including learning local language and culture, new schooling system, financial constraints, and sense of uncertainty and displacement/isolation affecting all family members [32]. In support of this view, it was recently found that children and adolescents of expatriate families had more severe COVID-19 related posttraumatic stress symptoms than children/adolescents of Saudi citizens which may negatively affect their e-learning [33].

## Strengths and limitations

This study has important strengths because it is one of few available adequate population-based studies for exploring the VLE satisfaction of both Saudi citizens and non-Saudi residents school students and their parents. The used questionnaire was checked after its implementation by all possible statistical means and it was found to be of adequate and acceptable validity and reliability to have robust/sound results and solid conclusions.

The limitations of this study included the lower participation from eastern, northern and southern regions of the kingdom due to the presence of the highest population densities in the central and western regions as well as the uneven distribution of online questionnaires which depended on social media and internet resources/accessibility and inability of the questionnaire to reveal specific details of the defects/problems in some aspects of the VLE including the role of the instructor and assessment evaluation. However, this online questionnaire was the only way to reach participants in view of the inability to directly approach participants in different regions due to lockdown measures. Moreover, thorough investigation of the various contributing factors that determine parent and child satisfaction including the necessary equipment/internet resources for VLE, quality of school and the form of e-learning could not be achieved. However, this questionnaire was designed as a screening tool to discover overall satisfaction and the main disadvantages/problems that emerged during the VLE experience. If a too long questionnaire was used, lower responses or more incomplete responses might be expected because the inclusion of more questions can result in a tedious questionnaire requiring more time and effort from participants. This study may not be nationally representative of parents of school students by education and employment status due to higher participation of employed fathers and mothers who had university and postgraduate higher education which can be attributed to their more resources for equipment and internet accessibility and a greater ability to respond and participate in this online survey. However, age and gender distribution of the studied sample was fairly close to demographics of population in Saudi Arabia. Also, it is practically very difficult or nearly impossible to map/collect sample which is consistent with all sociodemographic characteristics of the population through an online survey. No qualitative research, pilot study or face validity of the questionnaire was conducted because of difficulties encountered during quarantine period and lockdown measures. Additionally, test-retest reliability was not done which can be investigated in future studies. Finally, this survey was done during the first few months after school closure due to COVID-19 pandemic, when presumably, schools were still adjusting to e-learning.

## Conclusions

The participants in this survey were not very satisfied with the Saudi VLE experience and 62.8% of school children (62.8%) still preferred traditional education for many given reasons. However, fortunately, the Saudi VLE experience is better than no education or the lack of facilities for any VLE for populations who are digitally-excluded without internet access in resource-poor countries. The VLE was accepted as an alternative to traditional classrooms in the current COVID-19 pandemic to keep up with learning and to maintain the safety of children and it can be just a supplementary learning method. VLE technology can play a major role in helping school children learn and develop new skills because future education seems to be more digitally-dependent. It is recommended to support e-learning advancement in the future with successful VLE implementation at the school level whether during normal or emergency situations. Many measures are still needed to develop and improve the VLE, such as managing technical problems especially the free supply of equipment/internet, including

their cost in school fees, training stakeholders, developing more innovative and interesting teaching tools, and implementing a more synchronized VLE.

## Supporting information

**S1 File.**
(PDF)

**S1 Data.**
(ZIP)

## Author Contributions

**Conceptualization:** Moustafa Abdelaal Hegazi, Mohamed Hesham Sayed, Turki Saad Alahmadi, Mohamed Saad El-Baz, Mohammad Ahmed Altuwiriqi.

**Data curation:** Nadeem Shafique Butt, Nadeem Alam Zubairi, Mohamed Saad El-Baz, Ali Fahd Atwah, Mohammad Ahmed Altuwiriqi, Fajr Adel Saeedi, Nada Mansour Abdulhaq, Saleh Huwidi Almurashi.

**Formal analysis:** Nadeem Shafique Butt.

**Investigation:** Nada Mansour Abdulhaq.

**Methodology:** Moustafa Abdelaal Hegazi, Nadeem Shafique Butt, Mohamed Hesham Sayed, Turki Saad Alahmadi, Mohamed Saad El-Baz, Mohammad Ahmed Altuwiriqi.

**Project administration:** Nadeem Alam Zubairi.

**Resources:** Turki Saad Alahmadi, Fajr Adel Saeedi, Saleh Huwidi Almurashi.

**Software:** Nadeem Shafique Butt, Ali Fahd Atwah, Nada Mansour Abdulhaq, Saleh Huwidi Almurashi.

**Supervision:** Mohamed Hesham Sayed, Nadeem Alam Zubairi.

**Validation:** Moustafa Abdelaal Hegazi, Nadeem Shafique Butt, Mohamed Hesham Sayed, Nadeem Alam Zubairi, Turki Saad Alahmadi, Ali Fahd Atwah, Saleh Huwidi Almurashi.

**Visualization:** Mohamed Saad El-Baz, Ali Fahd Atwah, Saleh Huwidi Almurashi.

**Writing – original draft:** Moustafa Abdelaal Hegazi, Mohamed Hesham Sayed, Turki Saad Alahmadi, Mohammad Ahmed Altuwiriqi, Fajr Adel Saeedi.

**Writing – review & editing:** Nadeem Shafique Butt, Nadeem Alam Zubairi, Mohamed Saad El-Baz, Ali Fahd Atwah, Fajr Adel Saeedi, Nada Mansour Abdulhaq.

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
