## [Decision Letter · Decision Letter 0]

2 Aug 2022

PONE-D-22-14889Evaluation of the virtual learning environment by school students and their parents in Saudi Arabia during the COVID-19 pandemic after school closurePLOS ONE

Dear Dr. Butt,

Thank you for submitting your manuscript to PLOS ONE. After careful consideration, we feel that it has merit but does not fully meet PLOS ONE’s publication criteria as it currently stands. Therefore, we invite you to submit a revised version of the manuscript that addresses the points raised during the review process.

We look forward to receiving your revised manuscript.

Kind regards,

Patrick Charland

Academic Editor

PLOS ONE

Journal Requirements:

2. Please provide additional details regarding participant consent. In the Methods section, please ensure that you have specified (1) whether consent was informed and (2) what type you obtained (for instance, written or verbal). If your study included minors, state whether you obtained consent from parents or guardians. If the need for consent was waived by the ethics committee, please include this information.

3. Please include a copy of your questionnaire or scale as a Supporting Information file or provide a link if it is available through an online repository.

4. PLOS ONE does not copy edit accepted manuscripts (https://journals.plos.org/plosone/s/criteria-for-publication#loc-5). To that effect, please ensure that your submission is free of typos and grammatical errors.

Reviewers' comments:

Reviewer's Responses to Questions

**Comments to the Author**

1. Is the manuscript technically sound, and do the data support the conclusions?

Reviewer #1: Yes

Reviewer #2: Yes

2. Has the statistical analysis been performed appropriately and rigorously? 

Reviewer #1: I Don't Know

Reviewer #2: Yes

3. Have the authors made all data underlying the findings in their manuscript fully available?

Reviewer #1: Yes

Reviewer #2: Yes

4. Is the manuscript presented in an intelligible fashion and written in standard English?

Reviewer #1: Yes

Reviewer #2: Yes

5. Review Comments to the Author

Reviewer #1: The problematic of this study is well defined. The first few sentences of the article clearly demonstrate the field of education in which this article is situated. Moreover, the manuscript meets the objectives of the journal PLOS One advance science for the benefit of society. The problematic is well supported by varied and recent references, which is necessary for a text presenting information related to a rather new situation, the Covid-19. Thus, we congratulate the authors for this effort of documentary research.

The methodology chosen seems robust and takes into account the standards of ethical research involving human beings. Regarding the statistical analyses, I would like to mention to the editor that my expertise in advanced statistical analysis is limited. I therefore reserve comment on this aspect and confine myself to comments on the other elements of the article (coherence between sections, adequacy between the objectives and the theoretical framework, methodology, etc.).

The questionnaire used was constructed by the research team and is based on previous research. It was also verified by experts. The third section of the questionnaire sought to assess the learner's and parents' perspectives on their experience with the VLE. However, the lack of information on the theoretical basis for the satisfaction construct is questionable. We are aware that the article is succinct, but the theoretical foundations on which the questionnaire is based are missing, both for the definition of VLE and for satisfaction and comparison with the traditional classroom. We suggest that the author shorten considerably the first part of the discussion section, which is intended as a reminder of the problematic, in order to give the necessary space to the theories underlying the constructs of the questionnaire. Similarly, at the end of the manuscript, we are still unable to grasp what VLE actually involves. Several questions remain unanswered about this form of learning. Is VLE conducted by a trained teacher who was monitoring students prior to the health crisis or by other teachers, or parents? This lack of information makes it difficult to appreciate the discussion section, as many questions remain in the reader's mind. One avenue for refining the theoretical framework might be to properly define the VLE, but also traditional teaching and place it in the context of teaching in Saudi Arabia. This context may differ from other countries and a mapping of this context is necessary.

Having read about it, I'm not surprised at the low response rate of fathers in this study, knowing that the parenting role often falls to the mother. I'm glad this was mentioned in the article, it's important. In this regard, Table 1 is very clear and provides a picture of the participants in the study. We applaud the authors' efforts in presenting this table.

In the Results section, it is stated that "According to the definition of satisfaction, participants evaluated the VLE experience as unsatisfactory...". However, we are unable to locate this definition in the text. A section should be added to the theoretical framework.

The results regarding the difference in satisfaction between elementary and secondary school children are very interesting and bring new data to the field of educational research.

Limitations and avenues for future research are well identified at the end of the article.

Reviewer #2: Are you talking about a particular VLE or about distance education at large ? In the first case, some details about the VLE would have been appreciated. It would also have been interesting to know more about the pandemic in RSA (how many lockdowns ? how long?) and to compare your resultats with other researches made in other countries.

6. PLOS authors have the option to publish the peer review history of their article (what does this mean?). If published, this will include your full peer review and any attached files.

Reviewer #1: No

Reviewer #2: No

---

## [Author Response · Author response to Decision Letter 0]

28 Aug 2022

Reply to Comments of Reviewer (1):

1. The reviewer was concerned about the lack of information on the theoretical basis for the satisfaction construct, the theoretical foundations on which the questionnaire is based, and the definition of VLE and for satisfaction. 

Response/Reply: Regarding definition of VLE, in the introduction section, VLE was defined as remote learners receive their learning materials electronically via the internet (Zhang, Zhao, Zhou, & Nunamaker, 2004). However, more details about particular VLE conducted in Saudi Arabia was added and highlighted in the methods section under the subheading (Study design and participant selection).

Regarding the theoretical foundation on which the questionnaire was based, it was mentioned in the methods section (subheading: Measuring tool) that the items of the questionnaire suitable for school students were based on previous studies and expert opinions from validated instruments for assessing distance education learning environments in higher education, integrative review for barriers and solutions to online learning and literature reviews and simultaneously included the opinion of parents in VLE ((Zhang, Zhao, Zhou, & Nunamaker, 2004 & Walker & Fraser, 2005 & Strong, Irby, Wynn & McClure, 2012 & O'Doherty et al., 2018). 

In the methods section (Subheading: Statistical validation/reliability analysis of the questionnaire), it was mentioned that the used questionnaire was checked after its implementation by all possible statistical means and it was found to be of adequate and acceptable validity and reliability. Moreover, exploratory factor analysis was conducted for 24 variables (questions) using parallel analysis to determine the number of factors to retain with direct oblimin rotation. The following reference of another previous research applied the same method of statistical validation and reliability analysis and exploratory factor analysis used in our survey. 

Hamutoglu NB, Gemikonakli O, Duman I. Kirksekiz A, Kiyici M. Evaluating students experiences using a virtual learning environment: satisfaction and preferences. Education Tech Research Dev. 2020; 68: 437-462. 

According to exploratory factor analysis, the whole questionnaire consisted of 5 main domains: 

- Learner domains (3): (perspective on the VLE, skill and interaction, perspective on traditional classrooms) 

- Parent domains (2): (facilities/support and parental role/perspective on the VLE) 

Regarding the definition for participant satisfaction about VLE, in the methods section (Subheading: Statistical validation/reliability analysis of the questionnaire), it was mentioned that the answers to the questions were rated on a 5-point Likert scale from 1-5 (strongly disagree=1, disagree=2, true sometimes=3, agree=4 and strongly agree=5).

In the methods section (Subheading: Questionnaire implementation and distribution), it was mentioned that in this study, learner and parental VLE satisfaction was the main dependent or outcome variable. A score with a median value above 3 indicated overall satisfaction with the VLE experience as well as satisfaction with each individual domain, while a median value below 3 indicated dissatisfaction. However, this important paragraph was transferred and highlighted under a new subheading Definition of the main outcome (VLE satisfaction) to be more prominent. 

It is clear to choose “3” as an average score on five-point Likert scale. It is worth mentioning that we did not introduce any cut-off for satisfaction rather using “3” being the central category of the Likert scale to interpret the tendency of satisfaction scores.

The following reference adopted the same cutoff value and this reference was added to the reference list. The same value for measuring satisfaction to each question evaluating VLE was exactly the same value used in our survey The mean of all questions was calculated and all reported values above 3.0, the mid-point on the scale indicated satisfaction. 

Barker J and Gossman P. The learning impact of a virtual learning environment: students' views. Teacher Education Advancement Network Journal (TEAN). 2013; 5:19-38.

Reply to Comments of Reviewer (2):

The reviewer asked if authors are talking about a particular VLE or about distance education at large, to add some details about the particular VLE in KSA, to know more about the pandemic in KSA (how many lockdowns? how long?) and to compare results with other researches made in other countries.

Response/Reply: More details about particular VLE conducted in KSA during COVID-19 crisis and relation of study time to number and time of lockdowns were added and highlighted in the methods section under the subheading (Study design and participant selection).

Regarding comparing results with other researches made in other countries. Authors thank the reviewer for giving the opportunity to emphasize that up to the best of our knowledge, this study is unique because it is the first comprehensive study either worldwide or from KSA to consider the opinion of both school students and their parents simultaneously (not in higher education setting as done in other previous researches) in the evaluation of satisfaction for different aspects of VLE. 

However, it was mentioned in the discussion section that: To mitigate the spread of SARS-CoV-2 among children, considering that their safety is a priority, the governments of 188 countries introduced social distancing and lockdown measures with unprecedented worldwide closure of face-to-face child services such as schools interrupting education for >90% of the world’s students or 1.5 billion children and youth establishing a learning crisis (Human Rights Watch, 2020 & UNESCO b Education: From disruption to recovery, 2020 & Lee, 2020) While more than two-thirds of countries have introduced a national distance learning platform, only 30% of low-income countries have done so (UNESCO c, National learning platforms and tools, 2020 & Selbervik, 2020). The COVID-19 pandemic implies limited or no education for many children. Not only did school closure in many countries with planned extended lockdowns interrupted teaching but many exams have also been postponed, rescheduled or cancelled (Daniel, 2020 & UNESCO b, Education: From disruption to recovery, 2020).

Consequently, it was mentioned in the conclusion that: However, fortunately, the Saudi VLE experience is better than no education or the lack of facilities for any VLE for populations who are digitally-excluded without internet access in resource-poor countries.

---

## [Editor Report · Decision Letter 1]

12 Sep 2022

PONE-D-22-14889R1Evaluation of the virtual learning environment by school students and their parents in Saudi Arabia during the COVID-19 pandemic after school closurePLOS ONE

Dear Dr. Butt,

Thank you for submitting your manuscript to PLOS ONE. After careful consideration, we feel that it has merit but does not fully meet PLOS ONE’s publication criteria as it currently stands. Therefore, we invite you to submit a revised version of the manuscript that addresses the points raised during the review process.

We look forward to receiving your revised manuscript.

Kind regards,

Patrick Charland

Academic Editor

PLOS ONE

Journal Requirements:

Additional Editor Comments:

*** Prof Nadeem Shafique Butt,

In light of the external reviewers' comments, your rebuttal letter and the comments made on the manuscript, I am pleased to announce its "unofficial" acceptance.

However, you will note that I have placed the paper in "minor revision" mode, as some slight changes will need to be made to the reference section.

As mentioned in the submission instructions (https://journals.plos.org/plosone/s/submission-guidelines#loc-references)

- All references should be presented according to the ICMJE style (Vancouver).

- This style also implies naming the journals according to their official abbreviations which you can find here: https://www.ncbi.nlm.nih.gov/nlmcatalog/journals

As soon as I receive the revised manuscript in the light of these corrections to be made to the references I will be able to proceed towards the official acceptance.

Thank you for your collaboration,
---

## [Author Response · Author response to Decision Letter 1]

14 Sep 2022

Dear Editor,

Thank you very much for accepting our manuscript. The following issues have been addressed and a revised version of the manuscript has been uploaded to the submission portal.

- References are rechecked and presented according to ICMJE style (Vancouver).

- The references list uses official abbreviations to name journals (where applicable).

- All bibliographic items are managed using Endnote PloS style.

Regards,

Nadeem

---

## [Editor Report · Decision Letter 2]

15 Sep 2022

Evaluation of the virtual learning environment by school students and their parents in Saudi Arabia during the COVID-19 pandemic after school closure

PONE-D-22-14889R2

Dear Dr. Butt,

We’re pleased to inform you that your manuscript has been judged scientifically suitable for publication and will be formally accepted for publication once it meets all outstanding technical requirements.

Kind regards,

Patrick Charland

Academic Editor

PLOS ONE

---

## [Editor Report · Acceptance letter]

20 Sep 2022

PONE-D-22-14889R2 

Evaluation of the virtual learning environment by school students and their parents in Saudi Arabia during the COVID-19 pandemic after school closure 

Dear Dr. Butt:

I'm pleased to inform you that your manuscript has been deemed suitable for publication in PLOS ONE. Congratulations! Your manuscript is now with our production department. 

Kind regards, 

on behalf of

Dr. Patrick Charland 

Academic Editor

PLOS ONE